# Network study of miRNA regulating traumatic heterotopic ossification

Kun Lian[1], Zhiyan Chen[2], Leijie Chen[3], Yongmei Li[2]*, Luping Liu[3]*

**1** Department of Neurosurgery, The Second Affiliated Hospital of Kunming Medical University, Kunming, Yunnan, China, **2** Department of Rehabilitation Medicine, The Second Affiliated Hospital of Kunming Medical University, Kunming, Yunnan, China, **3** Department of Orthopedics, The Second Affiliated Hospital of Kunming Medical University, Kunming, Yunnan, China

☉ These authors contributed equally to this work.
* cherry2009666@163.com (YL); liuluping@126.com (LL)

**Data Availability Statement:** Because the data were obtained from patients, we did not upload the sequencing data to a public database, which was limited by hospital policies and patient privacy. Data can be obtained by contacting corresponding

## Abstract

### Objective

Objective: To identify and analyze the microRNAs that are expressed differently (DE-miR-NAs) and forecast their potential roles in the pathophysiological process of traumatic heterotopic ossification (THO).

### Methods

We conducted RNA sequencing on six samples of normal bone and THO tissues from the patients and conducted differential expression analysis of miRNA. The biological activities of the target genes of the differentially expressed microRNAs (DE-miRNAs) were investigated using Gene Ontology (GO) and Kyoto Encyclopedia of Genes and Genomes (KEGG) analyses. The miRNA-mRNA network was constructed using Cytoscape software, incorporating miRNAs with varying expression levels and their corresponding target genes.

### Results

In comparison to the normal control group, a total of 84 differentially expressed microRNAs (p<0.05, |log2FC|>1) were identified, with 27 microRNAs showing up-regulation and 57 microRNAs showing down-regulation. The functional enrichment analysis revealed that the target genes of the de-mirna were primarily enriched in biological processes such as the regulation of protein stability and the management of neuromuscular process balance. Additionally, a miRNA-mRNA expression regulatory network was established. The RT-qPCR analysis revealed that miR-142-3p, miR-150-5p, miR-421, miR-625-5p, miR-675-5p, and miR-940 exhibited a decrease in expression levels in THO tissues. Nevertheless, the expression levels of miR-181c-3p, miR-320c, miR-497-5p, and miR-99a-5p were increased in THO tissues.

### Conclusions

Our investigation has uncovered the expression patterns and projected the potential activities of differentially expressed microRNAs (DE-miRNAs) in human THO. This research may

author Professor Yongmei Li at cherry2009666@163.com. In addition, you can also obtain the original data from Professor Bao Zhu of the hospital Ethics Management Committee via his email address 842508364@qq.com.

**Funding:** This study was supported by a joint special grant from the Department of Science and Technology of Yunnan Province and Kunming Medical University (202001AY070001-066). In this study, the funders provided the necessary resources for the smooth conduct of the study by providing financial support. Funders supported the financial needs of the study, which may include experimental materials, equipment use, staff salaries, and other costs associated with the study.

**Competing interests:** The authors have declared that no competing interests exist.

contribute to a better understanding of the underlying mechanisms and offer new possibilities for therapeutic targets in THO.

## 1.Introduction

Traumatic heterotopic ossification (THO) is a debilitating condition characterized by the abnormal formation of bone in muscles and soft tissues. This condition is commonly caused by trauma (in up to 75% of cases), surgery (in up to 40% of cases), and other local or systemic injuries [1,2]. The clinical signs of THO include persistent pain, irreversible joint activity loss, nerve and blood vessel damage, and skin ulcers, leading to a decrease in quality of life [3]. For chronic symptomatic THO, surgical surgery is the only definite choice, however it comes with various hazards such as wound-healing problems and high recurrence rates [4,5]. Additional preventive strategies are currently restricted to radiation and non-steroidal anti-inflammatory medicines (NSAIDs), such as indomethacin. However, it is important to note that these interventions might potentially result in severe consequences. Furthermore, the specific molecular biology and genetic targets that these measures act upon are not yet fully understood [6,7]. Currently, the exact pathological process behind the production of traumatic heterotopic ossification is not well understood, which is impeding the progress in developing safe and efficient treatments.

miRNA is a set of small RNA molecules that do not code for proteins. They are composed of around 18–30 nucleotides and have a significant function in the process of posttranscriptional RNA silencing [8]. Previous studies have reported on the involvement of miRNA in the pathogenesis of skeletal illnesses such as osteoporosis, osteoarthritis, fracture, and heterotopic ossification [9]. The miRNA-mRNA regulatory axis significantly influences the regulation of osteogenic differentiation in stem cells from various sources throughout the initiation and progression of these illnesses. Studies have shown that miR-128-3p can enhance the process of marrow mesenchymal stem cells (MSCs) transforming into bone cells, which helps slow down the onset of osteoporosis. This is achieved via activating the Wnt/3a signaling pathway [10]. Xiong et al. discovered that miR-26a-5p has the ability to reduce the expression of phosphatase and tensin homolog (PTEN) and promote the healing of bone fractures in individuals with traumatic brain injury [11]. Qin et al. showed that suppressing miR-17-5p can decrease the production of abnormal bone in samples taken from the hip joint capsule of humans by targeting ANKH [12].

MiRNA plays a specific role in regulating and varying the expression of factors involved to ossification. Additionally, miRNA strictly controls the formation of THO. This study utilized high-throughput sequencing technology to analyze the expression patterns of miRNA in the THO lesions. Additionally, it aimed to predict target genes and provide annotations regarding their activities. Furthermore, we utilized GO and KEGG pathway analyses to anticipate the potential activities of the DE-miRNAs, and we also created miRNA-mRNA networks. The gene function of DE-miRNA was initially anticipated, and the miRNA-mRNA regulation network associated with osteogenesis was constructed. By conducting the aforementioned studies, we have conducted an initial investigation into the pathways through which miRNA and their target genes contribute to the development of THO. This research will aid in better understanding the underlying mechanisms of THO and serve as a foundation for identifying potential therapeutic targets.

## 2. Materials and methods

### 2.1 Sample description

From August 1, 2023 to February 1, 2024, a total of 14 patients who underwent surgical treatment in the Second Affiliated Hospital of Kunming Medical University were enrolled in this study, including 7 patients in THO group and 7 patients in control group. The study was approved by the Ethics Committee of the hospital (checked-PJ-2023-140), and written informed consent was obtained from all participants.The inclusion criteria for the experimental group consisted of individuals who underwent resection of THO and had radiologically verified significant THO surrounding the joint that was operated on. The control group consisted of patients who had undergone the removal of internal fixation and had no previous history of total hip arthroplasty. The exclusion criteria included failure to provide informed consent for the procedure or participation in the trial, as well as a diagnosis of osteochondroma or illness, such as respiratory or urinary tract infection.

The samples were obtained using 2ml cryogenic tubes and promptly preserved in liquid nitrogen at a temperature of -196°C for future use, following the addition of 2ml of RNAstore. Three samples were chosen from each group for miRNA sequencing, while the remaining samples were utilized for subsequent verification tests.

### 2.2 Small RNA sequencing

The miRNeasy Kit (QIAGEN, USA) was utilized to extract the whole of RNA from tissue samples, following the manufacturer's given technique. The RNA integrity was assessed using the Bioanalyzer 2100 (Agilent, CA, USA), and high-throughput sequencing was conducted following a thorough quality review.

sRNA libraries were prepared using the TruSeq sRNA Sample Prep Kits (Illumina, San Diego, USA) following the instructions provided with the kit. The precise protocols were as follows: The procedure involves 3' splice sequence ligation, 5' splice sequence ligation, reverse transcription to generate cDNA strand, PCR amplification, electrophoresis purification of the target fragment, and sequencing using a Hiseq2500 sequencer (Illumina, San Diego, USA) with 1x50bp read length. The methodology employed throughout the entire experiment is illustrated in Fig 1.

### 2.3 Screening of DE-miRNAs

Fold change filtering was applied to identify DE-miRNAs between THO lesions and normal bone tissues. Student's t-test was employed to evaluate the statistical significance of the difference. miRNAs with | log2 (fold change) |>1 and P<0.05 were selected as the significantly differentially expressed, miRNAs with log2 (fold change) > 2 were marked as upregulated, miRNA with log2 (fold change) <0.05 were marked as downregulated, and miRNAs that did not meet the above conditions were not significantly differentially expressed genes.

### 2.4 Real-time quantitative polymerase chain reaction (RT-qPCR)

Table 1 displays ten DE-miRNAs, with four being upregulated and six being downregulated, that were chosen to validate the accuracy of the sequencing data. The reverse transcription of miRNA was performed using the miRcute improved miRNA cDNA first-strand synthesis kit (TIANGEN) with the specified conditions:The temperature will be set at 42 degrees Celsius for a duration of 60 minutes, followed by a temperature of 95 degrees Celsius for a duration of 3 minutes. The mRNA levels were assessed by quantitative real-time PCR analysis (RT-qPCR) using the miRcute enhanced miRNA fluorescence quantitative detection kit (TIANGEN). The

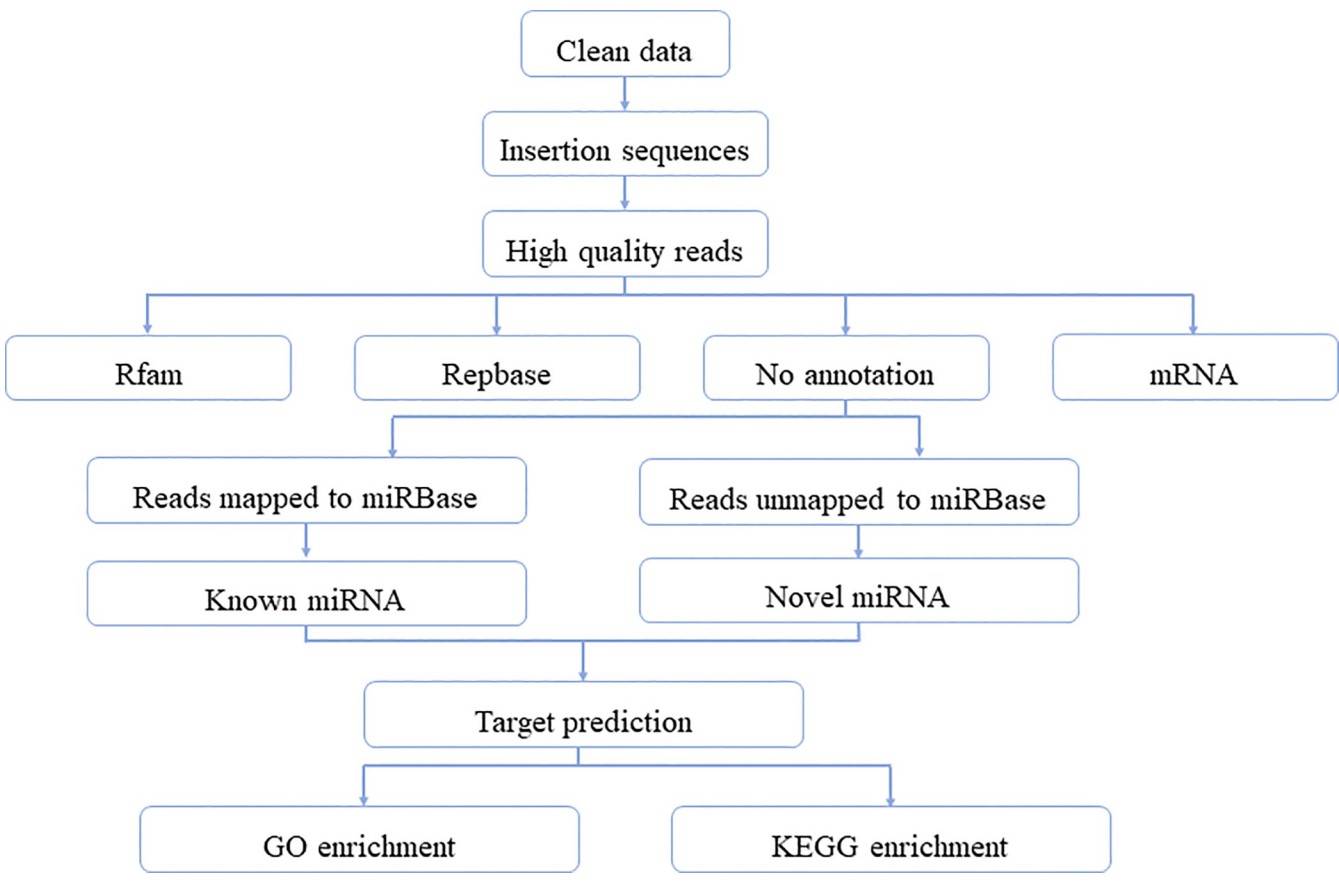

**Fig 1. Schematic diagram of the analytical procedures used in this study.**

RT-qPCR experiment was performed using a typical two-step PCR amplification program consisting of 45 cycles. Each cycle included a denaturation stage at 95°C for 15 minutes, followed by an annealing step at 94°C for 20 seconds, and an extension step at 60°C for 34 seconds. The ABI 7500 instrument from the USA was used for this experiment. The miRNA

**Table 1. The primer list used for real-time quantitative PCR.**

| Gene | Primer sequence (5'-3') |
|---|---|
| hsa-miR-142-3p | Forward: GGCTGTAGTGTTTCCTACTTTATGGA |
| hsa-miR-150-5p | Forward: TCTCCCAACCCTTGTACCAGTG |
| hsa-miR-181c-3p | Forward: AACCATCGACCGTTGAGTGG |
| hsa-miR-320c | Forward: CAAAAGCTGGGTTGAGAGGGT |
| hsa-miR-421 | Forward: ATCAACAGACATTAATTGGGCG |
| hsa-miR-497-5p | Forward: CAGCAGCACACTGTGGTTTGT |
| hsa-miR-625-5p | Forward: GCAGGGGGAAAGTTCTATAGTCC |
| hsa-miR-675-5p | Forward: TGGTGCGGAGAGGGCCCAC |
| hsa-miR-940 | Forward: AAGGCAGGGCCCCCGCT |
| hsa-miR-99a-5p | Forward: AACCCGTAGATCCGATCTTGTG |
| U6 | Forward: CTCGCTTCGGCAGCACA |
| | Reverse: AACGCTTCACGAATTTGCGT |

expression levels were determined using the 2-ΔΔCt technique, with U6 serving as a reference gene for normalization.Three separate experiments were conducted.

## 2.5 GO and KEGG analysis

For additional bioinformatics investigation, GO and KEGG pathway analyses were conducted to estimate the function of the target genes of miRNA. The prediction of miRNA/mRNA interaction was conducted using TargetScan 5.0 and miRanda 3.3a. The establishment of miRNA-mRNA regulatory networks was accomplished through the utilization of Cytoscape.

## 2.6 Statistical analysis

The statistical analyses were conducted using SPSS, version 26.0, developed by IBM Corp. in Armonk, NY, USA. The mean ± standard deviation (means ± SD) is used to express all important data. The student's t-test was utilized to assess the disparity between the two groups. The threshold for statistical significance was established at a P-value of less than 0.05. The data were graphically represented using GraphPad Prism 9 software (GraphPad Software, Inc., Bethesda, MD, USA).

# 3. Results

## 3.1 General information of the subjects

Table 2 displays the fundamental details of the individuals in both the THO group and the control group. Additionally, Fig 2 presents the X-ray scans of patients with THO, with one image picked for each site. The average age of the THO group was 35.25±10.5 years, and the average duration of the condition was 233±165.5 days. The average age of the control group was 43±10.8 years, and the average duration of the disease was 538.75±371.20 days.

## 3.2 DE-miRNAs between normal bone and THO tissues

A total of 84 differentially expressed microRNAs (DE-miRNAs) were identified in normal bone and THO tissues. The fold change (FC) values for these DE-miRNAs ranged from 0.1 to 14.9. Among them, 27 miRNAs were up-regulated while 57 miRNAs were down-regulated. The two most notable differentially expressed microRNAs were miR-877–5p (FC = 0.12, $p < 0.01$) and miR-4510 (FC = 10.74, $p < 0.01$). The miR-4492 gene had the highest fold rise (FC = 14.9, $p < 0.01$), while the PC-5p-13936 gene showed the most fold decrease (FC = 0.1, $p < 0.01$) in THO. The volcano plot and cluster heat map were employed to visually represent the differential expression of DE-miRNAs in various samples, respectively (Fig 3).

## 3.3 Validation of the sequencing data by RT-qPCR

Following an extensive screening of DE-miRNAs, we validated specific transcriptome disparities using RT-qPCR. We examined osteogenic factors consisting of four upregulated and six

**Table 2. Clinical data of participants.**

| Group | Age (years old) | Injury time (day) | Location | | | | |
|---|---|---|---|---|---|---|---|
| | | | Shoulder | Knee | Elbow | Hip | Ankle |
| S (n = 7) | 35.25±10.5 | 233±165.5 | 1 | 1 | 4 | 1 | / |
| D (n = 7) | 43±10.8 | 538.75±371.20 | / | 1 | 5 | / | 1 |

S: THO group; D: Control group.

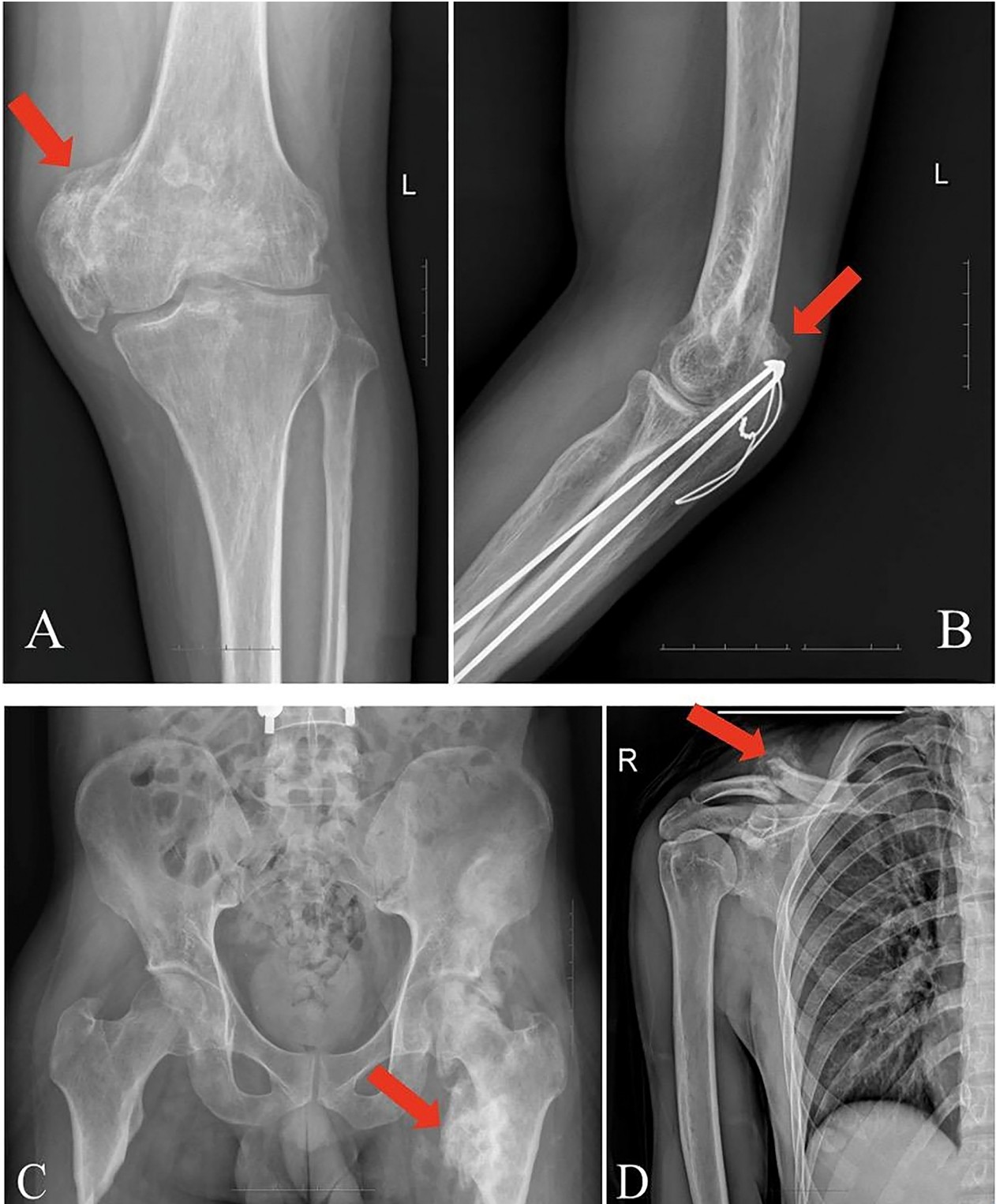

**Fig 2. X-ray of THO groups. The red arrows indicate traumatic area of heterotopic ossification.** (A) Patella; (B) Humerus bone; (C) The femoral; (D) Clavicle bone.

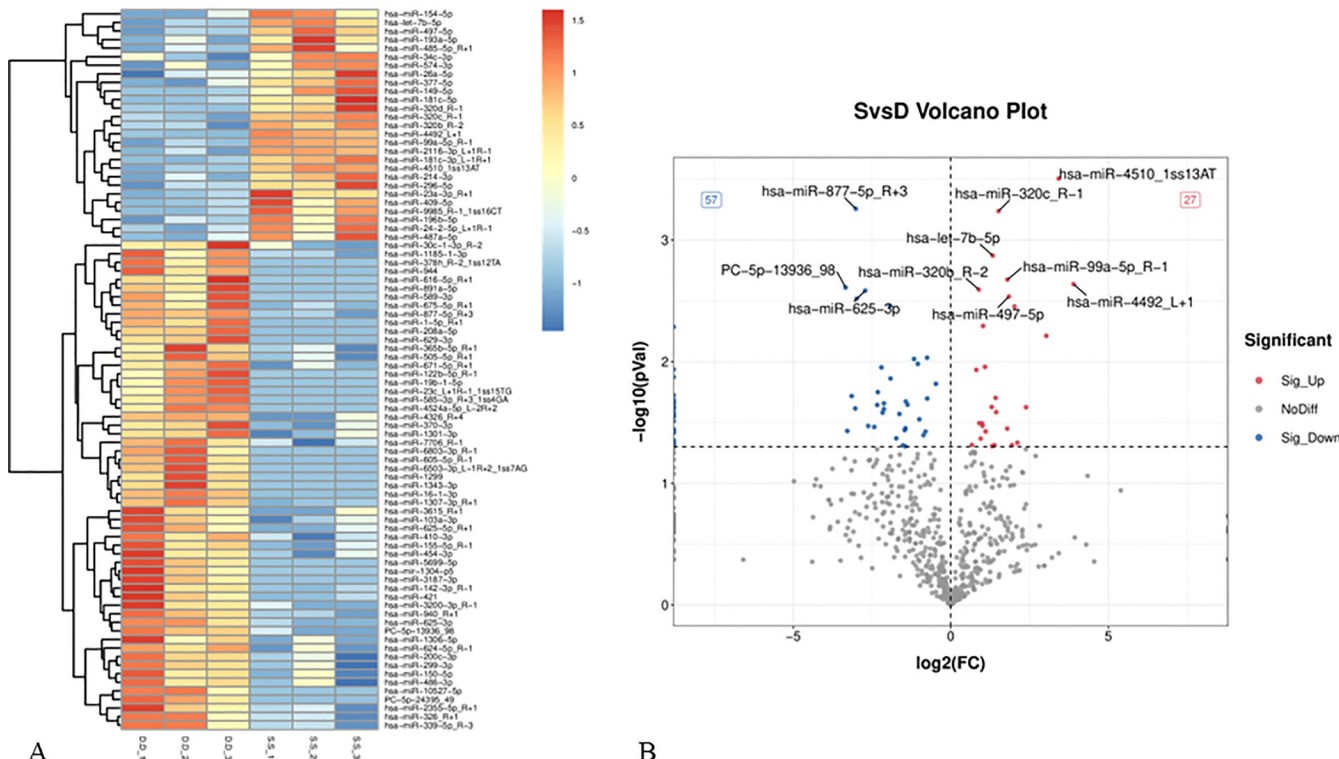

**Fig 3. Paired differential expression analyses between normal bone and THO tissues.** (A) The heatmap indicates the differentially expressed miRNA profiles in the six samples. "Red" represents the higher expression, while "blue" represents the lower expression level. (B) Volcano plot with the differentially expressed genes. Blue circles represent downregulated miRNAs. miRNA, microRNA; THO, traumatic heterotopic ossification.

downregulated miRNAs and performed RT-qPCR analysis on four THO tissue samples and four normal bone samples. The analysis revealed that the levels of miR-181c-3p, miR-497-5p, miR-99a-5, and miR-320c were significantly increased in THO tissues, while the levels of miR-142-3p, miR-625-5p, miR-940, miR-675-5p, miR-150, and miR-421 were significantly decreased in THO tissues. These findings are consistent with the sequencing data. Fig 4 displays the RT-qPCR validation results for the 10 differentially expressed miRNAs in this experiment.

## 3.4 GO and KEGG enrichment

The GO annotations identified genes that were highly elevated and downregulated in THO, in comparison to bone. The GO annotations were linked to various biological processes, molecular functions, or cellular components. The most important Gene Ontology (GO) terms in THO were regulation of protein stability, cell leading edge, and SMAD binding, as shown in Fig 5A. The KEGG pathway analysis identified a limited number of pathways that were either overrepresented or underrepresented in THO. THO exhibited an increase in cancer pathways, specifically the Rap1 signaling route, Axon guidance, Ras signaling pathway, and PI3K-Akt signaling pathway, as compared to normal bone (Fig 5B).

## 3.5 THO-specific miRNA interactome and hub genes

We utilized 6 samples that had both miRNA and mRNA sequencing data in order to construct a THO-specific miRNA interactome consisting of 10 differentially expressed (DE) miRNAs. Fig 6 illustrates the network, which has two prominent clusters of interconnected miRNA-mRNA pairs. One cluster involves downregulated miRNAs (green), whereas the second cluster

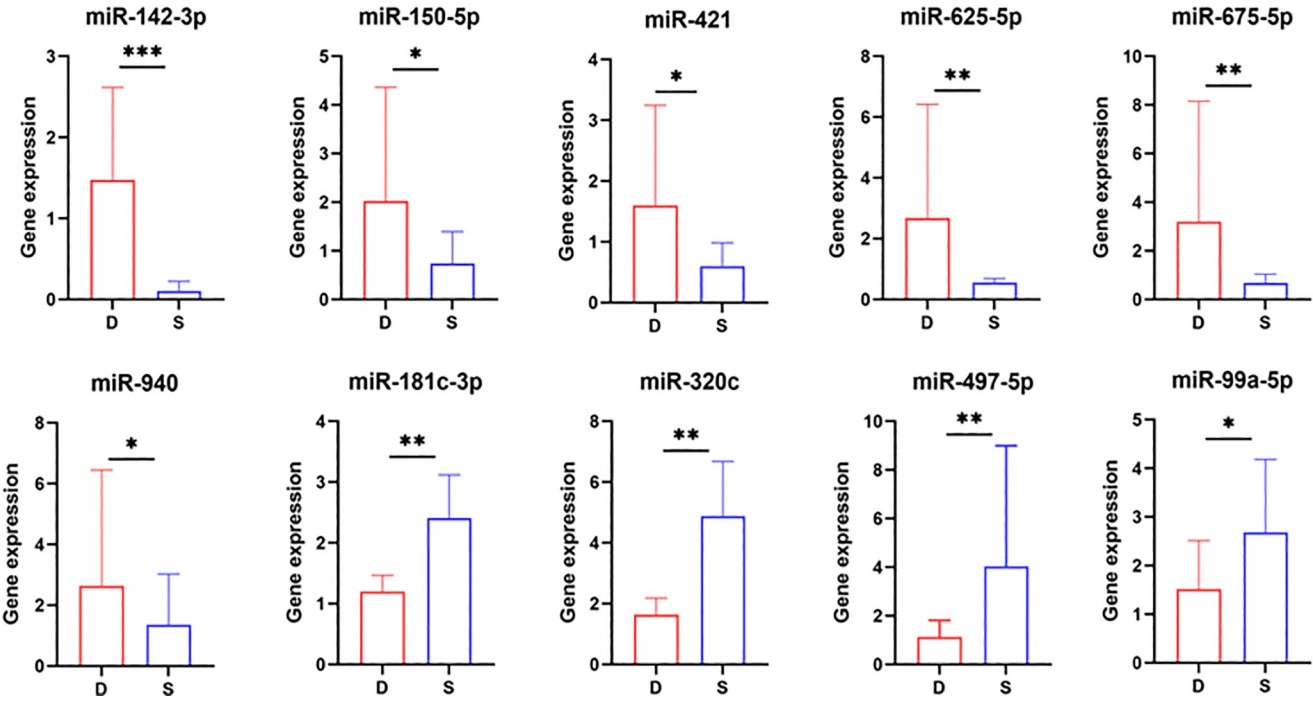

**Fig 4. RT- qPCR analysis of miRNA levels in bone (D) and THO (S) samples.**

involves upregulated miRNAs (red).The analysis revealed that miR-142-3p, miR-150-5p, miR-421, miR-625-5p, miR-675-5p, miR-940, miR-181c-3p, miR-320c, miR-497-5p, and miR-99a-5p have a significant impact on the interaction between DE-mRNA and DE-miRNA.The network proposed that miRNAs have the ability to regulate target genes by competitively binding to mRNA.

## 3.6 Validation of the target genes by WB

After analyzing the gene prediction findings, we identified WDR12, PDE1C, and HDAC11 as possible targets of miR-150-5p, miR-625-5p, and miR-940, respectively (Fig 7). By doing a

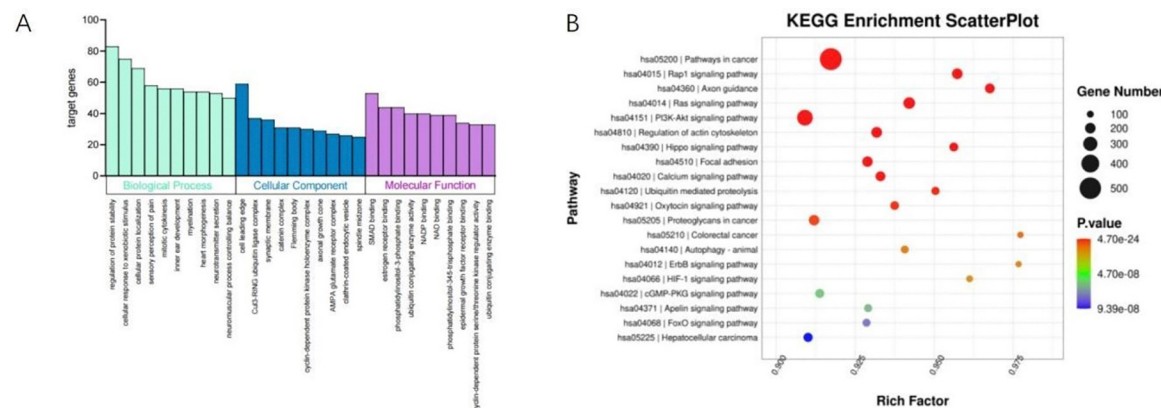

**Fig 5. GO and KEGG analyses for DE-miRNAs.** (A) GO analysis predicted the enriched genes in biological processes, cellular components or molecular functions based on their fold enrichment scores. (B) KEGG pathway analysis predicted the enriched pathways of DE-miRNAs parental genes.

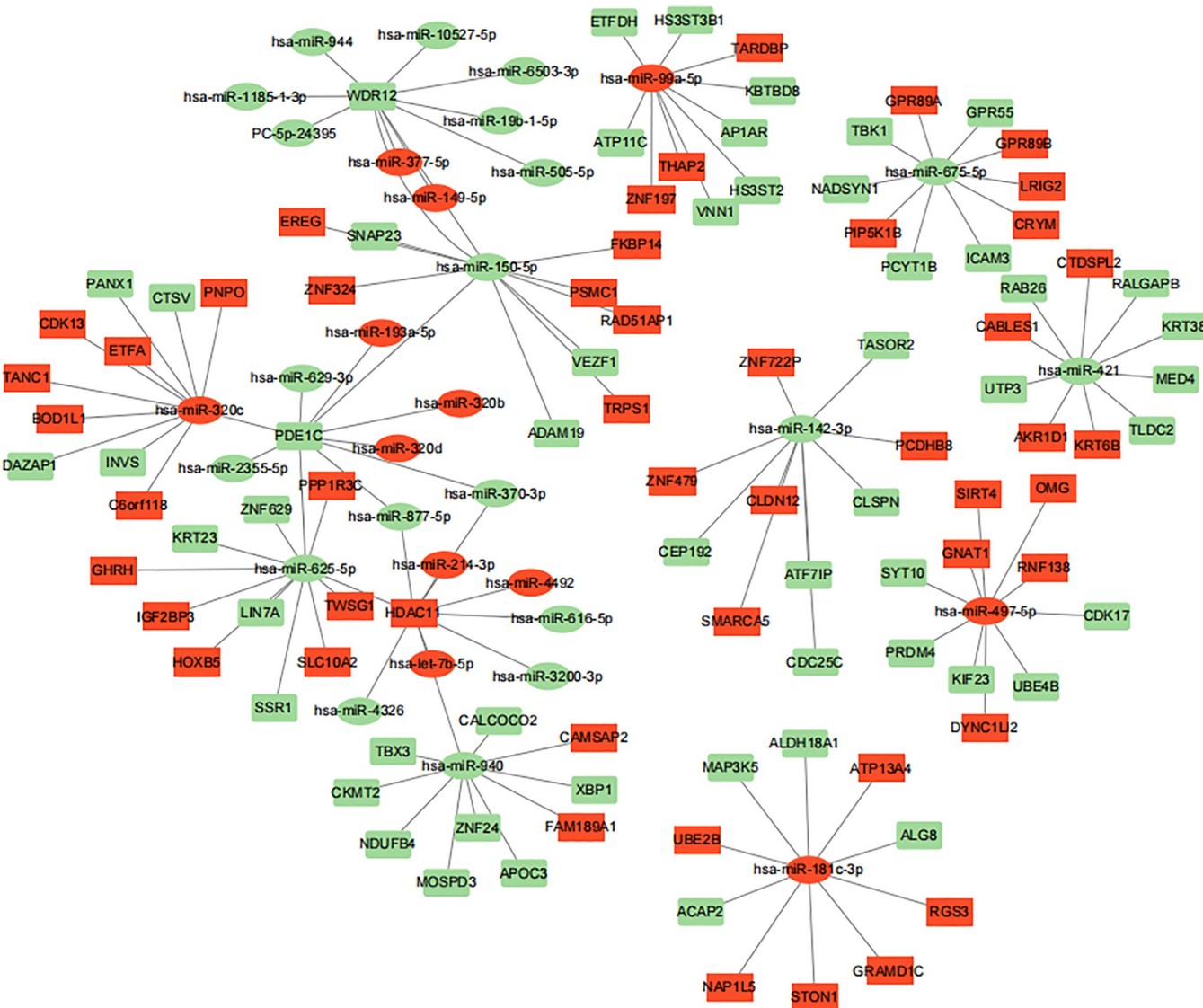

**Fig 6. THO miRNA–mRNA interactome.** Network of differentially expressed miRNAs targeting differentially expressed genes between normal and THO tissues.the rectangle is mRNA, the oval is miRNA, and the red and green denote the differentially up-regulated and down-regulated gene, respectively, edges denote that a miRNA targets the connected gene. miRNA, microRNA; mRNA, messenger RNA; THO, traumatic heterotopic ossification.

literature search, we discovered that these three genes exhibited a favorable correlation with osteogenic differentiation. While the target genes of differentially expressed microRNAs (DE-miRNAs) have been discovered, their specific function in THO (Thyroid Hormone Overdose) remains ambiguous. Thus, we chose clinically diagnosed THO tissues and normal bone tissues to analyze the protein expression levels of WDR12, PDE1C, and HDAC11 using Western blotting. The findings indicated that THO tissues exhibited high expression of WDR12, PDE1C, and HDAC11 proteins (Fig 8).

## 4. Discussion

Given the high occurrence rate, poor quality of life, and resulting handicap associated with THO, numerous research has aimed to uncover the primary variables through investigating

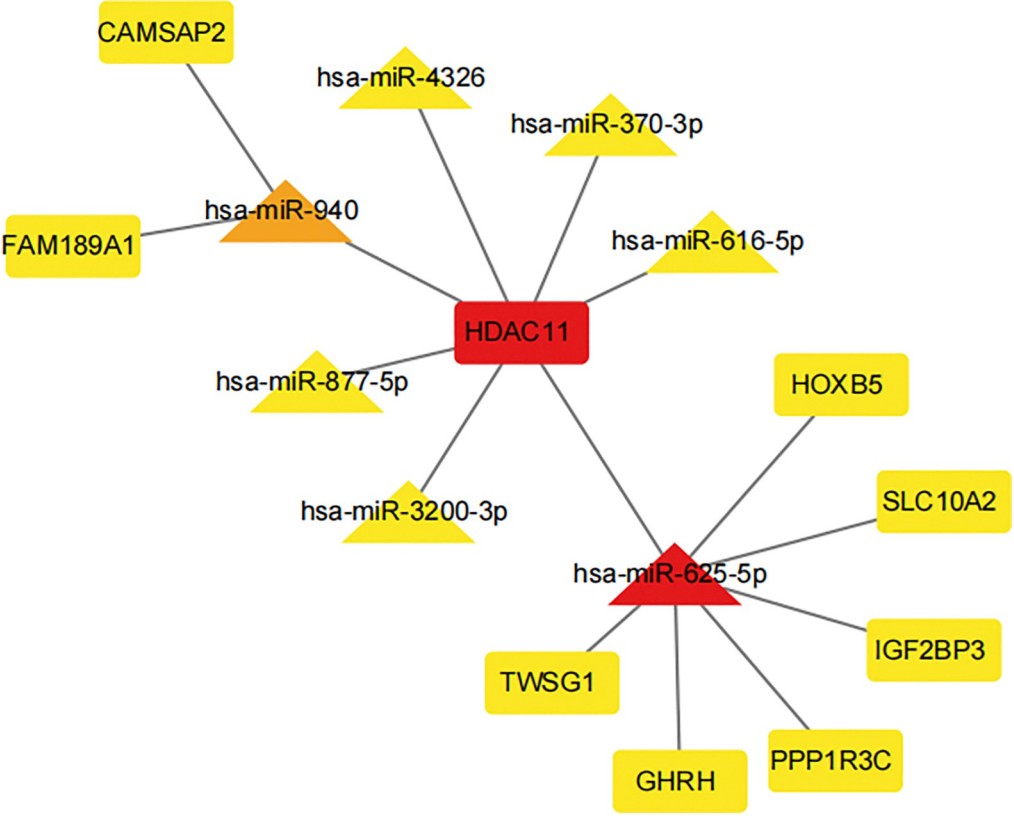

**Fig 7. Hub miRNA and Hub mRNA.**

the injury mechanism, wound treatment, complications, and subsequent THO formation [2]. Nevertheless, the intricate molecular biological process remains incompletely comprehended, and the management of THO is limited to NSAIDs, radiation, and surgical excision [13]. Hence, the examination of pathogenic processes, clinical presentation, diagnosis, therapy, and prevention is of utmost importance and immediate necessity. By analyzing the miRNA profiles in THO lesions, identifying the differentially expressed miRNAs using bioinformatics methods, and investigating their mechanisms in THO, we can offer new theoretical and experimental support for diagnosing and treating THO using miRNA.

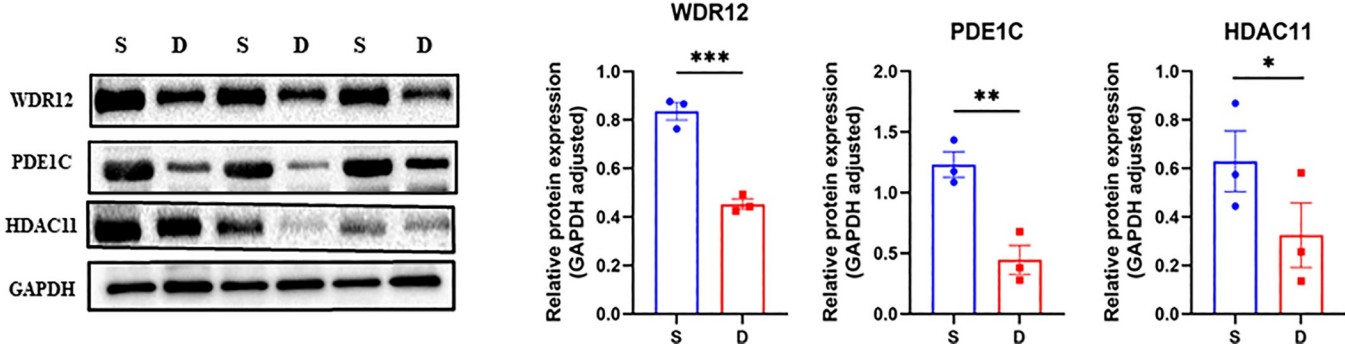

**Fig 8. Quantification of protein level of WDR12, PDE1C, HDAC11 normalized to GAPDH.**

## 4.1 Analysis of DE-miRNA expression in THO tissues

The present study employed RNA sequencing to uncover a total of 27 upregulated miRNAs and 57 downregulated miRNAs in THO lesions. The miRNAs were selected based on two criteria: a $|log2FC|>1$ and $p<0.05$. Among the up-regulated differentially DE-miRNAs, miR-4510 exhibited the greatest significance, with a FC of 10.74 and $p <0.01$. Conversely, miR-4492 exhibited the highest FC = 14.9 with $p <0.01$. However, among the down-regulated differentially expressed miRNAs, the most statistically significant one was miR-877-5p, with a FC of 0.12 and $p <0.01$. Furthermore, PC-5p-13936 demonstrated the most significant decrease in FC = 0.1 and $p <0.01$. Only a limited subset of these DE-miRNAs has been recently documented in the context of THO. The miRNAs that were found to be significantly differentially expressed in this study are miR-142-3p (FC = 0.1, p-value < 0.05), miR-421 (FC = 0.36, $p < 0.05$), and miR-574-3p (FC = 2.01, $p < 0.05$). Most of the other DE-miRNAs have not been studied in relation to THO. In a controlled laboratory setting, the upregulation of miR-574-3p impeded the differentiation of mesenchymal stem cells (MSC) into chondrocytes. Furthermore, the up-regulation and down-regulation of miR-574-3p both impeded the process of cartilage and bone development in the THO model in vivo. These findings suggest that miR-574-3p may have a significant impact on maintaining the specific characteristics of chondrocytes and the ability of MSCs to differentiate into several cell types [14]. In addition, certain members of separate miRNA families showed varying levels of expression in this study, such as miR-320 and miR-625 families, both of which contribute, to some extent, to the process of osteogenic differentiation. During hypoxia, miR-320-3p promotes osteogenic differentiation of MC3T3-E1 cells by inducing inflammation, apoptosis, and the body's response to oxidative stress. This is accomplished by triggering the activation of particular proteins, namely RUNX2, ALP, and OCN [15].

Furthermore, we performed further GO and KEGG pathway analyses to functionally identify the putative target genes of miRNAs. The GO analysis revealed a high enrichment of genes involved in protein stability control and localization, cell response, SMAD protein binding, estrogen receptor binding, and phosphatidylinositol 3-phosphate binding. The SMAD protein plays a crucial role in the creation of THO, hence facilitating the incidence of THO. During inflammation, the expression of SMAD protein is increased, leading to the activation of the SMAD and NF-κB signaling pathway and the initiation of THO production. MAD 5, the primary component in the SMAD signaling pathway, typically serves as an indicator of the level of activity of this pathway. The expression of SMAD 5 may suggest the creation of THO [16]. The development of THO involves the activation of osteogenic signaling pathways in order to stimulate the differentiation of THO progenitor cells into both chondrocytes and osteoblasts. Transcripts associated with tumor, Rap 1 signaling, Ras, PI3K-Akt, calcium, and HIF 1 signaling pathways were shown to be elevated in THO as compared to normal bone. The Rap1, PI3K-Akt, and HIF 1 signaling pathways are linked to osteogenic differentiation [17]. Additionally, the PI3K-Akt and HIF 1 signaling pathways are directly connected to the production of THO [18]. Dong et al. hypothesized that the activation of the PI3K-AKT signaling pathway is a crucial requirement for the development of THO triggered by BMP 2. They anticipate that targeting this pathway could be a potential approach for treating THO [19]. The HIF 1 signaling pathway triggers the production of THO following a burn injury by causing alterations in blood vessels and endothelial cells [20]. This study specifically examined 10 differentially expressed microRNAs (DE-miRNAs) in THO tissues, which are known to be associated with osteogenesis.

The authors conducted a study where they analyzed the expression profiles of mRNA, miRNA, and lncRNA in THO tissue, muscle, and normal bone tissue from patients undergoing total hip replacement. They discovered that certain miRNAs, namely miR-99b, miR-146,

miR-204, miR-195, and miR-143, which are involved in angiogenesis, as well as miR-148, which is associated with chronic inflammation, were found to be up-regulated in THO tissues [21]. Furthermore, in this study, it was found that miR-615 and miR-132 had a negative effect on the process of bone formation and the differentiation of bone cells in a laboratory setting. They achieved this by directly affecting the expression of two important regulators of bone formation, FOXO1 and GDF5, respectively. On the other hand, miR-563 was found to be considerably elevated in patients with OPLL. miR-10a controls the process of bone formation and the development of abnormal bone in cells of the posterior longitudinal ligament in living organisms [22]. Nevertheless, miR-99b exhibited consistency with the findings of this investigation, while the remaining miRNAs did not display any notable variations. However, we have discovered other miRNAs that have not been documented in THO. Further investigation is required to understand their roles in THO.

## 4.2 DE-miRNAs associated with osteogenic differentiation

Among the up-regulated DE-miRNAs, miR-181c-3p, miR-320c, miR-497-5p, and miR-99a-5p were all found to be involved in the process of osteogenic differentiation through distinct pathways. miR-181c-3p, miR-320c, and miR-99a-5p have the ability to control the process of bone marrow MSCs differentiating into bone cells. However, there is a distinction between them. The increase in miR-181c-3p enhances the osteogenic differentiation of BMSCs by activating the Wnt/β-catenin signaling pathway [23]. On the other hand, the increase in miR-320c and miR-99a-5p hinders the potential of BMSCs to differentiate into bone cells by targeting Runx2 [24,25]. The activation of miR-497-5p can stimulate the SMAD and Wnt/β-catenin signaling pathways, which are crucial for THO development, by targeting various molecules. For instance, miR-497-5p specifically targets RSPO2 in order to activate the Wnt/β-catenin pathway, which leads to the acceleration of ossification in the posterior longitudinal ligament [26]. Additionally, Smurf2 targets are activated to stimulate the SMAD signaling pathway, which promotes the osteogenic or odontogenic differentiation of human papilla stem cells [27]. Overexpression of miR-181c-5p and miR-497-5p in individuals with osteoporosis leads to increased osteoblast differentiation and mineralization, which is linked to the gradual loss of bone caused by the condition [28]. While these DE-miRNAs have not been identified in THO-related research, they have a role in osteogenic differentiation and are strongly associated with certain bone metabolic disorders. Hence, their participation in THO holds significant research significance.

During hypoxia, the levels of HIF-1α increase and it controls the activity of various genes including BMP, vascular endothelial growth factor, and neuropilin-1. This regulation influences biological processes such as angiogenesis, osteogenesis, and bone resorption. It also promotes the transformation of osteogenic precursor cells into chondrocytes and the development of abnormal bone tissue. This abnormal bone tissue plays a crucial role in the pathological formation of THO bones [29]. The upregulation of miR-675-5p controls the process of cell differentiation into bone cells by simultaneously triggering the HIF-1α response and activating the Wnt/β-catenin signaling pathway [30]. The present investigation found a strong association between cancer and the control of HIF-1 by miR-625-5p, miR-150-5p, miR-421, miR-625-5p, miR-675-5p, miR-940, and miR-142-3p. Suppression of miR-150-5p can enhance the growth of chondrocytes, trigger cell death, and break down the extracellular matrix when exposed to IL-1β [31]. Additionally, miR-150-5p exosomes stimulate the PI3K/AKT pathway via PTEN to facilitate the healing of skin wounds [32]. In the context of osteoporosis, miR-940 and PI3K/AKT were identified as the central components of the mirNA-gene interaction network and KEGG enriched pathways, respectively. Furthermore, miR-940 demonstrated favorable performance in the ROC curve analysis for osteoporosis [33].

Research indicates that miR-421 and miR-142-3p play a role in the genesis of THO. The alterations in miR-421 expression levels in bone tissue and blood of patients with humeral fracture following surgery indicate the healing of the fracture and the potential for THO. Within a time, frame of 8 to 14 days following surgery, patients with heterotopic ossification experienced an increase in the levels of BMP-2 in both bone tissue and blood. Conversely, the levels of miR-421 were considerably reduced in comparison to patients without heterotopic ossification [34]. miR-142-3p, which is a kind of miRNA that inhibits bone formation, was shown to be highly expressed 2 and 4 days after muscle injury. This led to the development of neurogenic HO. However, when there was an injury to the central nervous system, the osteoinhibitory impact of miR-142-3p was no longer present [35].

## 4.3 Mechanisms by which miRNAs regulate key genes to mediate THO

Upon confirming the presence of these de-mirnas in THO tissues, we integrated the sequencing results with a database search to identify the target genes of these DE-miRNAs. Through this process, we identified critical genes involved in osteogenic differentiation, including WDR12, HDAC11, and PDE1C.

WD repeat domain 12 (WDR12), a gene associated with cancer, serves essential roles in the pathway of ribosome synthesis. Its removal greatly hinders cell proliferation [36]. A prior study demonstrated that lung adenocarcinoma cells located near the ossification site produced bone morphogenetic protein-2 and osteopontin, which typically promote and enhance bone growth [37]. Another study shown that the excessive expression of BMP-2 stimulates the formation of heterotopic ossification in rectal cancer [38]. Another study demonstrated that the histogenesis of HO in this particular case of metastatic colon cancer originates from the stromal cells present in the tumor microenvironment [39]. HO is present in various types of malignancies, suggesting that HO may have a similar molecular biology pathway with cancer. A prior investigation demonstrated that the expression of WDR12 undergoes considerable alterations in human HO lesions, hence playing a crucial role in bone development [2].

PDE1C, which is controlled by miR-625-5p, is a significant constituent of the phosphodiesterase 1 (PDE1) superfamily. It possesses the ability to interact with both cAMP and cGMP as substrates. PDE1C can alter the tumor microenvironment and trigger the development of colorectal cancer through the gene polymorphism of the calcium signaling pathway. It has a role in the immune invasion of colorectal cancer. The downregulation of PDE1C in the immune microenvironment suppresses immune infiltration, whereas the upregulation of PDE1C results in increased CD8+T cell infiltration, heightened inflammatory response, elevated immune score, and enhanced activity of PD-L1, CD27, and CD40. These findings suggest that targeting PDE1C could potentially enhance personalized immunotherapy for colorectal cancer [40]. Upregulating the expression of PDE1C weakens the influence of histone demethylase on the epigenetic regulation of adipose-derived MSCs, leading to enhanced cell proliferation and differentiation [41]. Therefore, we postulated that miR-625-5p might influence THO by means of PDE1C's impact on the immunological microenvironment.

Osteoclasts play a crucial role in maintaining healthy bones by facilitating bone remodeling and aiding in the healing of fractures. Nevertheless, excessive osteoclast activity can result in bone disorders such as osteoporosis. HDAC11, a member of class IV HDAC, is expressed during the late stage of osteoclast development and is regulated by miR-940 [42]. The absence of HDAC11 expression in osteoclasts results in heightened osteoclast differentiation and activity. Conversely, the suppression of HDAC11 encourages the development of osteoclasts, indicating that HDAC11 can function as a suppressor of osteoclast differentiation [43]. Exposure to caffeine during pregnancy can lead to the harmful effects of abnormal growth in the long

bones of the unborn child, causing a condition known as fetal bone dysplasia. Caffeine consumption during adulthood decreased peak bone mass and heightened vulnerability to osteoporosis in males, but not females. Concurrently, the presence of caffeine decreased the expression of H3K9ac and 11β-hydroxysteroid dehydrogenase 2 (11β-HSD2) in male offspring both before and after birth but had no effect on female offspring. Furthermore, caffeine exposure leads to elevated levels of corticosterone, which in turn causes HDAC11 to bind to the promoter of 11β-HSD2. This binding is facilitated by the translocation of glucocorticoid receptors into the nucleus of BMSC. Additionally, it intensifies the suppressive impact of corticosterone on the process of osteogenic differentiation of bone marrow stromal cells (BMSC), resulting in the development of osteoporosis [44]. These findings indicate that miR-940 likely controls the activity of osteoclasts by influencing HDAC11 and has a role in the initiation and progression of THO.

## 4.4 Limitations and prospects

In this study, we identified 84 DE-miRNAs that were differentially expressed in THO lesions compared to normal bone tissues. Although our findings provide valuable insights into the molecular mechanisms of THO, we acknowledge some limitations. First, the sample size of this study was relatively small, with three randomly selected from the seven samples collected as a representative patient population for miRNA sequencing, which may limit the representability of our findings. Differences between individuals may affect the generalizability of our results. To address this issue, we used Western blot to verify the expression of key proteins associated with differentially expressed mirnas. Future studies with larger sample sizes and stratified sampling will be essential to further validate our results and explain differences in patient populations. In addition, case-control studies have had limited sample sizes due to the challenges of collecting THO cases, and the distinction between THO tissue and various types of soft tissue has not been studied. The results of this study demonstrate the critical importance of distinguishing the molecular features of THO from normal soft tissue and help elucidate the specific role of DE-miRNAs in THO pathogenesis. Therefore, animal models of THO and cell cultures from THO tissues and normal bone tissues are recommended for future studies to validate our findings and explore the mechanisms by which these mirnas regulate osteogenic differentiation. These experimental approaches could be useful based on our findings of mirna targeting THO key pathways such as BMP, HIF 1 and PI3K/AKT. With further studies, we can deepen our understanding of the underlying mechanisms of THO and provide directions for the development of new therapeutic interventions.

## 5. Conclusion

In comparison to regular bone tissues, there were 27 up-regulated and 57 down-regulated DE-miRNAs in THO lesions. DE-miRNAs can modulate the onset and progression of THO by interacting with osteogenesis-related proteins, including WDR12, PDE1C, HDAC11, and participating in osteogenesis-related pathways such as BMP, HIF 1, PI3K/AKT. The miRNA expression profile following trauma is linked to the onset and progression of THO. The miRNA-mRNA regulatory network has the potential to reveal new insights into the underlying mechanisms and identify promising treatment targets for THO.

## Supporting information

**S1 Raw images. S1 Raw images for the original base map of the original blot data.**
(PDF)

## Acknowledgments

The authors would like to thank all subjects who participated in this study. In particular, we thank Bioinfo_composer, the leading bioinformatics team in China, for the selfless help.

## Author Contributions

**Conceptualization:** Kun Lian, Luping Liu.

**Data curation:** Zhiyan Chen, Leijie Chen.

**Formal analysis:** Zhiyan Chen.

**Methodology:** Zhiyan Chen, Leijie Chen.

**Project administration:** Luping Liu.

**Resources:** Luping Liu.

**Software:** Leijie Chen.

**Supervision:** Yongmei Li.

**Writing – original draft:** Kun Lian.

**Writing – review & editing:** Yongmei Li.

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
