## [Decision Letter · Decision Letter 0]

24 Sep 2024

PONE-D-24-32971Network study of miRNA regulating traumatic heterotopic ossificationPLOS ONE

Dear Dr. Lian,

Thank you for submitting your manuscript to PLOS ONE. After careful consideration, we feel that it has merit but does not fully meet PLOS ONE’s publication criteria as it currently stands. Therefore, we invite you to submit a revised version of the manuscript that addresses the points raised during the review process.

We look forward to receiving your revised manuscript.

Kind regards,

Selvaraj Vimalraj

Academic Editor

PLOS ONE

“This study was supported by a joint special grant from the Department of Science and Technology of Yunnan Province and Kunming Medical University（202001AY070001-066).”

Additional Editor Comments:

revise the manuscript as per the reviewers suggestion.

Reviewers' comments:

Reviewer's Responses to Questions

**Comments to the Author**

1. Is the manuscript technically sound, and do the data support the conclusions?

Reviewer #1: Yes

2. Has the statistical analysis been performed appropriately and rigorously? 

Reviewer #1: Yes

3. Have the authors made all data underlying the findings in their manuscript fully available?

Reviewer #1: Yes

4. Is the manuscript presented in an intelligible fashion and written in standard English?

Reviewer #1: Yes

5. Review Comments to the Author

Reviewer #1: The study, conducted with meticulous attention to detail, aims to identify specific microRNAs differentially expressed in traumatic heterotopic ossification compared to normal bone formation. Tissue samples from THO and normal group patients were collected. RNAseq was used to identify differentially expressed miRNAs followed by GO and KEGG to understand target pathways and further validated by RT-PCR. The results clearly show that 87 DE RNAs were obtained (27 upregulated and 57 downregulated), and how they may be implicated in the development and therapeutic purpose are also discussed.

This thorough approach is a testament to the authors' expertise. The manuscript is well-structured, reflecting the authors' systematic approach to the investigation. However, a few points could be further clarified to ensure the manuscript is accessible to a wider audience. Those have been attached as a separate word file.

6. PLOS authors have the option to publish the peer review history of their article (what does this mean?). If published, this will include your full peer review and any attached files.

Reviewer #1: No

---

## [Author Response · Author response to Decision Letter 0]

29 Dec 2024

Dear editor,

Thank you very much for your warm work and sincere help. According to the reviewer's comments and the request of the journal, we have made some modifications as follows:

1.We adapted and revised the format in accordance with the style requirements of the journal.

2.Financial disclosure: In this study, the funders provided the necessary resources for the smooth conduct of the study by providing financial support. Funders supported the financial needs of the study, which may include experimental materials, equipment use, staff salaries, and other costs associated with the study.

3.Because the data were obtained from patients, we did not upload the sequencing data to a public database, which was limited by hospital policies and patient privacy. The miRNA sequencing data can be provided to you if needed. Data can be obtained by contacting corresponding author Professor Yongmei Li at cherry2009666@163.com. The data was collected and organized by Professor Li.In addition, you can also obtain the original data from Professor Bao Zhu of the hospital Ethics Management Committee via his email address 842508364@qq.com.4.We removed ethical statements from other sections.

5.The original base map for the original blot data is referred to S1_raw_images.pdf, which we supplemented in the manuscript.

6.We review our reference list to ensure that it is complete and correct.

Thank you again from the bottom of my heart. If you have any questions, please do not hesitate to contact me.

Best regards,

Lian

Dear Reviewer,

Thank you very much for your hard work and sincere guidance, and we sincerely appreciate your review of our manuscript.

Based on your valuable comments, we have revised the discussion section of the article and responded fully to the specific content, hoping that our modification can better compound the requirements of the article publication. If you have any unclear questions, please do not hesitate to contact me, we will reply in time.

Thank you again for your review.

Best wishes, 

Lian

Response to Suggestion 1:

We thank the reviewers for their interest in the representativeness of our sample size. We acknowledge that differences between individual patients may affect the representativeness of our study group. To address this, as you suggest, we included additional analyses performed using alternative methods, such as Western blotting, to validate our findings. This approach provides a more comprehensive validation beyond the scope of RNA sequencing, thus strengthening our conclusions. 

We added the following text to the "Discussion" section to reflect this: Given sample size limitations, we recognize the potential impact of patient differences on the results. To alleviate this problem, we used Western blotting to verify the expression of key proteins associated with differentially expressed mirnas. This additional layer of validation not only strengthens our findings but also demonstrates our commitment to addressing potential study limitations. Future studies with larger sample sizes and stratified sampling will be essential to further validate our results and explain differences in patient populations."

Response to Suggestion 2:

We understand the importance of comparing THO with soft tissues to understand the specificity of the miRNAs to the pathology. We have added the following explanation in the "Discussion" section to clarify the rationale behind this comparison:

"Comparing THO tissues with soft tissues is crucial for understanding the specificity of the miRNAs involved in the pathological process of heterotopic ossification. This comparison helps to differentiate the molecular signatures unique to THO from those present in normal soft tissues, which is essential for identifying potential therapeutic targets and biomarkers specific to THO."

Response to Suggestion 3:

We agree that discussing potential future experimental approaches would be beneficial for the readers. We have expanded the "Discussion" section to include a section on future directions, which outlines potential in vivo and in vitro models for validating our predictive model results and discusses the implications for therapeutic intervention development. The added text is as follows:

"The current study lays the foundation for future research by identifying differentially expressed miRNAs in THO. Future experimental approaches may include the use of various in vivo models, such as animal models of THO, to validate the role of these miRNAs in the development and progression of the disease. Additionally, in vitro studies using cell cultures derived from THO tissues and normal bone tissues could provide further insights into the mechanisms by which these miRNAs regulate osteogenic differentiation. These approaches will be crucial for translating our findings into potential therapeutic interventions, which could include the development of miRNA-based therapies targeting key pathways implicated in THO."

Based on your valuable guidance and in combination with the results of our research, we have revised Discussion Section 4.4 Limitations and Prospects 377-396 as follows:

In this study, we identified 84 DE-miRNAs that were differentially expressed in THO lesions compared to normal bone tissues. Although our findings provide valuable insights into the molecular mechanisms of THO, we acknowledge some limitations. First, the sample size of this study was relatively small, with three randomly selected from the seven samples collected as a representative patient population for miRNA sequencing, which may limit the representability of our findings. Differences between individuals may affect the generalizability of our results. To address this issue, we used Western blot to verify the expression of key proteins associated with differentially expressed mirnas. Future studies with larger sample sizes and stratified sampling will be essential to further validate our results and explain differences in patient populations. In addition, case-control studies have had limited sample sizes due to the challenges of collecting THO cases, and the distinction between THO tissue and various types of soft tissue has not been studied. The results of this study demonstrate the critical importance of distinguishing the molecular features of THO from normal soft tissue and help elucidate the specific role of DE-miRNAs in THO pathogenesis. Therefore, animal models of THO and cell cultures from THO tissues and normal bone tissues are recommended for future studies to validate our findings and explore the mechanisms by which these mirnas regulate osteogenic differentiation. These experimental approaches could be useful based on our findings of mirna targeting THO key pathways such as BMP, HIF 1, and PI3K/AKT. With further studies, we can deepen our understanding of the underlying mechanisms of THO and provide directions for the development of new therapeutic interventions.

---

## [Decision Letter · Decision Letter 1]

22 Jan 2025

Network study of miRNA regulating traumatic heterotopic ossification

PONE-D-24-32971R1

Dear Dr. Lian,

We’re pleased to inform you that your manuscript has been judged scientifically suitable for publication and will be formally accepted for publication once it meets all outstanding technical requirements.

Kind regards,

Selvaraj Vimalraj

Academic Editor

PLOS ONE

Additional Editor Comments (optional):

-

Reviewers' comments:

Reviewer's Responses to Questions

**Comments to the Author**

1. If the authors have adequately addressed your comments raised in a previous round of review and you feel that this manuscript is now acceptable for publication, you may indicate that here to bypass the “Comments to the Author” section, enter your conflict of interest statement in the “Confidential to Editor” section, and submit your "Accept" recommendation.

Reviewer #1: All comments have been addressed

2. Is the manuscript technically sound, and do the data support the conclusions?

Reviewer #1: Yes

3. Has the statistical analysis been performed appropriately and rigorously? 

Reviewer #1: Yes

4. Have the authors made all data underlying the findings in their manuscript fully available?

Reviewer #1: Yes

5. Is the manuscript presented in an intelligible fashion and written in standard English?

Reviewer #1: Yes

6. Review Comments to the Author

Reviewer #1: (No Response)

7. PLOS authors have the option to publish the peer review history of their article (what does this mean?). If published, this will include your full peer review and any attached files.

Reviewer #1: No

---

## [Editor Report · Acceptance letter]

24 Jan 2025

PONE-D-24-32971R1 

PLOS ONE

Dear Dr. Lian, 

I'm pleased to inform you that your manuscript has been deemed suitable for publication in PLOS ONE. Congratulations! Your manuscript is now being handed over to our production team.

Kind regards, 

on behalf of

Dr. Selvaraj Vimalraj 

Academic Editor

PLOS ONE